# Body Emotional Investment and Emotion Dysregulation in a Sample of Adolescents with Gender Dysphoria Seeking Sex Reassignment

**DOI:** 10.3390/jcm11123314

**Published:** 2022-06-09

**Authors:** Maria Giuseppina Petruzzelli, Lucia Margari, Flora Furente, Lucia Marzulli, Francesco Maria Piarulli, Anna Margari, Sara Ivagnes, Elisabetta Lavorato, Emilia Matera

**Affiliations:** 1Department of Basic Medical Sciences, Neuroscience and Sensory Organs, University Hospital “A. Moro”, Piazza Giulio Cesare 11, 70100 Bari, Italy; maria.petruzzelli@uniba.it (M.G.P.); lucia.marzulli@uniba.it (L.M.); piarullif@gmail.com (F.M.P.); margarianna2@gmail.com (A.M.); 2Department of Biomedical Sciences and Human Oncology, University Hospital “A. Moro”, Piazza Giulio Cesare 11, 70100 Bari, Italy; lucia.margari@uniba.it (L.M.); saraivagnes@gmail.com (S.I.); emilia.matera@uniba.it (E.M.); 3Psychiatry Unit, Azienda Ospedaliero-Universitaria Policlinico di Bari, 70100 Bari, Italy; e.lavorato@alice.it

**Keywords:** adolescents, gender dysphoria, transgender, emotion dysregulation, body investment, protection, internalizing symptoms, depression, self-harm, suicide

## Abstract

Adolescents with gender dysphoria (GD) often have internalizing symptoms, but the relationship with affective bodily investment and emotion dysregulation is actually under-investigated. The aims of this study are: (1) the comparison of Self-Administrated Psychiatric Scales for Children and Adolescents’ (SAFA), Body Investment Scale’s (BIS), and Difficulties in Emotion Regulation Scale’s (DERS) scores between GD adolescents (*n* = 30) and cisgenders (*n* = 30), (2) finding correlations between body investment and emotion regulation in the GD sample, (3) evaluating the link between these dimensions and internalizing symptomatology of GD adolescents. In addition to the significant impairment in emotion regulation and a negative body investment in the GD sample, Spearman’s correlation analyses showed a relationship between worse body protection and impaired emotion regulation, and binary logistic regressions of these dimensions on each SAFA domain evidenced that they may have a role in the increased probability of pathological scores for depression. Our results focused on the role played by emotion regulation and emotional investment in the body in the exacerbating and maintenance of internalizing symptoms, in particular depression, and self-harming behaviors in GD adolescents.

## 1. Introduction

Gender identity refers to an individual’s identification as male, female, or, occasionally, some category other than male or female [1]. The term “Gender Variance” includes a wide spectrum of gender experiences and behaviors pointing to a partial or complete mismatch between an individual’s gender identity and the sex established at birth [2]. “Gender Dysphoria” (GD) is a general descriptive term that refers to an individual’s affective/cognitive discontent with the assigned gender; meanwhile, in a diagnostic manner, according to the *Diagnostic and Statistical Manual of Mental Disorders 5th ed. (DSM-5),* it refers to the distress that may accompany the incongruence between one’s experienced or expressed gender and one’s assigned gender, resulting in significant psychological distress and impairment in important areas of functioning [3]. This discrepancy is the central component of the *DSM-5’s* Criteria A for GD, and it is described differently in children and adolescent/adults: children’s specifiers are more concrete and behavior-related than adolescent/adult ones, in which the subjective experience of gender and somatic sexual characteristics is the prominent part of the diagnosis. Criteria B is the same for both children and adolescents/adults, and it focuses on clinically significant distress and global impairment in several domains of functioning [3,4,5].

Recent epidemiological data on young individuals with GD [6,7] suggest that the number of adolescents referred to specialized gender identity clinics appears to be increasing, with heterogeneous data from different countries [8,9]. A large Netherlands cohort study about older subjects reported a prevalence of 1:11,900 male-assigned at birth and 1:30,400 female-assigned at birth in 1990 vs. 1:2800 male-assigned at birth and 1:5200 female-assigned at birth in 2015 [10]. Meanwhile, mostly in adolescence, it is evident that there is a progressive change in the sex ratio, from a higher proportion of male-assigned at birth to higher rates of female-assigned at birth [1,11,12,13], as shown in a study about sex ratio data in which, comparing years 1988–2006 and 2007–2016 in an Amsterdam Clinic, the sex ratio changed significantly, favoring female-assigned at birth (percentage of male-assigned at birth for the 2 time periods: 69.7% vs. 46.8%) [14]. Moreover, retrospective studies demonstrate that GD continues from childhood into adulthood in the range of 12–27%, suggesting that not all children and youths perceiving gender identities different from the biological sex would have persistent GD in their maturity [15]. A growing body of research reveals that GD individuals aged between 10 and 17 years old experience a range of psychiatric symptoms at rates higher than youths of the general population; in fact, Connolly et al. in 2016 [16] synthetized in a review the results of different studies of psychiatric comorbidities in adolescents with gender dysphoria, finding presence of depression in 12–64%, suicide attempts in 9–19%, self-harm in 13–46%, and eating disorders in 5–15%; moreover, it was found that there was a significative difference (*p* < 0.0001) in comparisons with cisgenders for the presence of depression and history of suicide attempts; lastly, this review also included two studies that compared psychiatric symptoms for gender dysphoria adolescents who had socially transitioned or who had been treated with steroid suppression, finding no difference with cisgenders of the same age [16,17]. It is well-established that adolescence is a period marked by the onset of various mental disorders [18,19], and that the impairment in self-perceived “body experience” has a strong association with a number of psychopathological conditions with gender-related clinical manifestations [20,21,22]. So, we argued that the inner process that leads to awareness of gender incongruity is accompanied by a negative emotional investment in one’s body as well as a broader difficulty regulating and managing negative emotions, and that both of these dimensions may play a significant role in the onset of internalizing symptoms.

Body investment is a multidimensional construct that refers to the cognitive, behavioral, and emotional importance of the body in one’s self-evaluation, and it theorizes that having a positive relationship with one’s own body increases one’s tendency for life preservation and attraction to life and serves as a shield against self-destruction [23]. As a result, it was expected that facets of body investment (body image, body care, body protection, and body touch) [23,24] would play a role in the prediction of non-suicidal self-harming injuries (NSSI) and acquired capability for suicide and suicide attempts [23,25]. In the meantime, the current literature focused attention on the well-established association between self-harming and self-destructive behaviors and greater degrees of emotion dysregulation [26,27,28]. Emotion regulation refers to a complex array of processes and strategies for monitoring, evaluating, and adjusting emotional experiences in the short-term and dynamically over time to accomplish goals [29]. These processes can be intrapersonal, arising from the inside of a person, and/or interpersonal, and a central task of adolescence is to learn how to control emotions in adaptive ways to promote social functioning and psychological well-being [30]. Although body investment is supposed to be associated with self-harming and self-destructive tendencies, and poor emotion regulation has been observed among children and adolescents with a variety of diagnoses, to our knowledge these dimensions have still not been investigated in GD adolescents.

Thus, the first objective of the current study was to compare: (a) internalizing symptoms, (b) emotional investment in the body, and (c) emotional regulation ability among a sample of adolescents with GD seeking sex reassignment (SR) with a comparison group of volunteers, without feelings of gender incongruences (cisgender) and without present or past formal psychopathological disorders, in the same age range. The second objective was to study the correlation between body investment and the ability to regulate emotions in the group of adolescents with GD; the last purpose was to evaluate if negative body investment and/or emotional dysregulation might be used to predict various aspects of internalizing symptomatology.

## 2. Materials and Methods

### 2.1. Subjects’ Recruitment

We enrolled a sample of individuals aged 12 to 18 years old at their first request of SR, who were referred to the Child and Adolescent Neuropsychiatry Unit of University of Bari “Aldo Moro” over a period of 24 months (April 2019–April 2021). The subjects were included in the study if they met the diagnostic criteria for GD according to *DSM-5* [2] after a specialist evaluation by physicians and psychologists from the Child and Adolescent Neuropsychiatry Unit and/or Service for Gender Dysphoria of Psychiatry Unit. Formal diagnosis of intellectual disability (ID) and autism spectrum disorder (ASD) were considered exclusion criteria for enrollment.

All participants and their parents underwent a clinical global assessment that looked at: (1) the reason for referral, (2) the onset timing and signs suggestive of GD in childhood or adolescence, (3) the history of neurodevelopmental disorders (according to *DSM-5*, with the exclusion of ID and ASD), (4) social behavior and relationships, (5) previous and current psychiatric symptoms including suicidal and self-injury behaviors. Cisgender volunteers included in the comparison group were recruited in two territorial high schools, after being informed about the aims of the study and stating their and their parents’ formal consent. They were asked to complete the assessment questionnaires if they were in the same range of age of the clinical group and with no previous or current history of formal psychopathological disorders. The study was approved by the Ethics Committee of the Policlinics of Bari (ED-AG).

### 2.2. Assessment of Psychopathology

The following structured self-report questionnaires were administered to the GD sample to complete the assessment of psychopathology and to the volunteer sample for comparison.

#### 2.2.1. Self-Administrated Psychiatric Scales for Children and Adolescents (SAFA)

SAFA is an Italian self-administered battery standardized on 895 Italian school children and adolescents and on 125 patients affected by different psychiatric disturbances, with age-specific scales for subjects aged 8 to 18 years old [31]. It investigates the internalizing dimensions of anxiety-related areas (SAFA A), depression-related areas (SAFA D), obsessive-compulsive symptoms (SAFA O), psychogenic eating disorders (SAFA P), somatic symptoms, and hypochondria (SAFA S) [32]. The raw scores obtained for each scale are converted to standardized (T) scores using age and sex reference tables (T = 50 + 10Z) and T scores differentiate pathological from non-pathological results according to the threshold T > 60. The psychometric properties of the questionnaire demonstrated adequate internal stability and consistency, with a Cronbach’s alpha > 0.80, and good one week test-retest stability (*p* < 0.01). Moreover, SAFA A, SAFA D, and SAFA P showed high convergent validity with other wide-validation questionnaires [33,34].

#### 2.2.2. Difficulties in Emotion Regulation Scale (DERS)

DERS is a 36-item self-report questionnaire [35], validated in Italian [36,37] and on adolescent populations [38,39]. It assesses six relevant domains of emotion regulation abilities: (1) non-acceptance of emotional responses (6 items), (2) difficulties engaging in goal-directed behavior (5 items), (3) impulse control difficulties (8 items), (4) lack of emotional awareness (5 items), (5) limited access to emotion regulation strategies (6 items), and (6) lack of emotional clarity (5 items). The sum of the scores of the six scales determines the overall score. High scores correspond to more difficulty in emotion regulation. Sighinolfi et al. 2010 [36] revealed strong internal consistency for the overall score of the Italian version, with Cronbach’s alpha of 0.90, and the six subscales exhibited values between 0.74 and 0.88 which were satisfactory too; moreover, Giromini et al. 2012 found adequate, and comparable to previous findings, internal consistency and test-retest reliability and good validity as indicated by concurrent validity analysis and comparison between a clinical and a non-clinical sample [37].

#### 2.2.3. Body Investment Scale (BIS)

BIS is a 24-item self-report [24], widely translated and diffusely used in scientific literature [40,41]. The scale is used in this paper in a self-made Italian translation due to the lack of an official version validated in this country. It examines the individual’s emotional investment in their own body, through the inclination toward conducting self-harm behaviors in the following four-factor domains with 6 items for each domain: (1) body care, (2) comfort in physical touch, (3) body protection, (4) image feelings and attitudes about the body [40]. The scale showed adequate consistency for each scale in the original version (Cronbach’s alpha ranging from 0.80 to 0.95) [23,24]; moreover, studies of validation on adolescent samples provided additional support for the four-factor solution and good capacity in differentiating the responses of suicidal and non-suicidal adolescents [42].

### 2.3. Statistical Analyses

All of the variables were recorded in a structured form specifically for this research. IBM SPSS Statistics 27 (SPS S.r.l.; Bologna, Italy) [43] was used to perform the analyses. Descriptive analyses were produced for sociodemographic characteristics including frequencies, means, and standard deviations. Assumptions of normality were tested using the Shapiro–Wilk test given the sample size. The psychometric parameters were compared between the clinical sample and the comparison group using the Mann–Whitney test for independent samples. To study the correlations between body investment feelings and emotional dysregulation in the clinical sample, Spearman’s (rs) coefficients were examined among BIS-C, BIS-T, BIS-P, BIS-I, and DERS-TOT and all of its scales. Moreover, univariate logistic binary regressions were performed to estimate the predictive individual capacity of each BIS domain and DERS-TOT as independent variables and each SAFA scale as dependent variable in both groups, using T = 60 as cut-off for pathological values, according to its psychometric proprieties. The assumption of non-collinearity was tested through the calculation of variance inflation factors (VIFs) [44]. The level of significance was set at *p* < 0.05.

## 3. Results

The total number of adolescents who came to our attention with a request for sex reassignment during the recruitment period was 33 subjects. Two of them were excluded because they had a diagnosis of autism spectrum disorder; moreover, one adolescent dropped out prior to undergoing psychometric assessment. The final GD sample of 30 subjects consisted of 23 female-assigned at birth and 7 male-assigned at birth with a mean age of 15.6 ± 1.6 (15.4 ± 1.64 for female-assigned at birth and 16.3 ± 1.25 for male-assigned at birth); the comparison sample included 26 females and 4 males, with a mean age of 16.3 ± 1 years for both males and females. The difference in sex ratio between the groups was tested by X^2^ test for categorical variables with no significance found (*p* = 0.317). Clinical characteristics of the GD sample obtained during the assessment are listed in Table 1.

Table 2 shows the comparison between the clinical and cisgender groups. In the GD sample, statistically significant higher scores were found in SAFA-A, SAFA-D, and SAFA-S, indicating the presence of pathological anxiety, depression, and somatic symptoms; otherwise, statistically significant lower scores were found in BIS-C, BIS-P, and BIS-I, indicating a negative attitude and feelings toward one’s own body, with the exception of discomfort in touch; moreover, with the exception of the impulse and awareness scales, statistically significant higher scores were reported for the total and all subscales of the DERS, implying a more severe emotional dysregulation in the clinical group.

For the GD sample, the correlations between the body investment domains and difficulties in emotion regulation are reported in Table 3. BIS-P is inversely associated with DERS-TOT and the awareness, strategies, and clarity scales; moreover, clarity has a statistically significant and negative correlation with BIS-C too.

Lastly, Table 4 summarizes the findings of the univariate binary logistic regression analyses performed on each SAFA scale in the GD and cisgender samples. According to the Hosmer and Lemeshow test, all analyses were significant. DERS-TOT emerged to predict each area of internalizing psychopathology evaluated by SAFA; otherwise, BIS-P emerged to be the only body investment factor that operates as a predictor for depressive and eating behavior symptomatology.

## 4. Discussion

In this study, we examined cognitive and affective processes related to body investment attitudes and emotion regulation abilities in a clinical sample of GD adolescents at their first request of SR.

The main finding was that adolescents with GD, when compared to a comparison group of cisgender volunteers, had significant impairment of both emotion regulation and emotional investment in the body, along with the presence of internalizing symptoms of anxiety, depression, and somatization. Moreover, we found that in the GD sample, a worse protective attitude toward the body and impaired emotion regulation abilities are interrelated and may have a link with the greater likelihood of pathological scores for depressive symptomatology and with symptoms of psychogenic eating disorders (in a weaker manner than with depression).

Previous studies conducted in different clinical populations have suggested that body image dissatisfaction, specifically, may play a role in the onset of depressive symptoms [45] and that emotion dysregulation acts as a transdiagnostic factor of vulnerability for multiple psychiatric disorders [46,47,48,49,50]. Despite this, to the best of our knowledge, the literature lacks specific data on emotion dysregulation and body emotional investment, more than just body image dislike, in GD adolescents [51].

We know that internalizing and externalizing symptoms are more common during the adolescent years, because of the complicated interaction between neurocognitive developmental processes and social pressures [52,53]. Thus, it is reasonable to assume that adolescents experiencing gender incongruities are exposed to a greater emotional burden, both because of the increased social pressures they face and due to the complex maturation of emotional regulation skills, closely linked to body investment.

Starting from the idea that the intensification of gender incongruence awareness is inherently distressing, we explored the hypothesis that negative affective investment in one’s own body as well as emotion dysregulation are intimately connected to gender incongruence, and that both these dimensions may have a role in the risk of internalizing symptoms in GD adolescents.

As expected, our results confirmed the data of the literature about a significant presence of internalizing symptoms (anxiety, depression, and somatic) and emotional difficulties in the GD group as compared to the cisgender volunteers’ sample [12,54,55]. However, the most clinically relevant data were found studying the correlation between BIS and DERS scores in GD adolescents, and their respective potential implication in the modulation of internalizing symptomatology. Indeed, we observed an inversely proportional correlation between the body protection score and different DERS subscales, suggesting a bidirectional link between low protective attitudes toward their own body and higher difficulty in emotion regulation in GD adolescents.

In the meantime, it is worthy to note that, evaluating the predictive role of each BIS domain and DERS total score on internalizing symptomatology (Safa A, D, O, P, S), the body protection domain has a strong specific relationship with depression and eating disorders symptomatology, while DERS total score is linked with all of the psychopathological dimensions examined [29]. As a result, we can argue that having a negative protective attitude about one’s body has a greater impact on depression than physical dislike of one’s appearance [56,57].

Globally considered, these findings suggest that adolescents with gender incongruence may be faced with difficult-to-manage negative emotions and depressive symptoms [40] when they have a bad protective attitude towards the body, rather than when they have difficulties in other domains of the multidimensional construct of BIS. This is of considerable clinical importance and requires certain therapeutic attention, especially during adolescent years, because the coexistence of emotion dysregulation, bad body protective attitude, and depressive symptoms may increase the risk of self-injurious and suicidal behavior.

Despite this, the study has some important factors that require attention; the main limitation is the small sample size, which reduces statistical power and limits the possibility of using more complex statistical analyses; moreover, the cross-sectional nature of the study does not allow one to make any causal inferences, therefore, larger sample sizes and longitudinal designs are needed to better understand the role of body investment and emotion dysregulation as a link between the intensification of gender incongruence awareness and internalizing symptomatology. Furthermore, attention should be paid to the way the recruitment was conducted: we excluded ID because it could compromise the validity of the scores of the questionnaires and ASD because of the frequent overlapping between gender dysphoria and autism that could represent a confounding factor [58], and because the way in which emotion regulation and emotional understanding act in ASD patients has different mechanisms and diverse awareness and produces different behaviors in these patients; however, the exclusion of this subjects limits the possibility of studying GD in this population. Lastly, the exiguity of multiple sources of information due to the retrospective method of data collection does not allow one to analyze eventual confounding factors regarding clinical and anamnestic data.

However, a key aspect of this study is the fact that it analyzes specific and understudied outcomes, such as body investment attitudes and emotion dysregulation among adolescents with gender dysphoria, a highly vulnerable but also not so extensively studied population. Furthermore, this work sheds new light on the idea that the distress typical of these individuals may be due to intrinsic suffering related to increasing awareness of gender incongruence, in addition to the issue of victimization and societal pressure.

## 5. Conclusions

In conclusion, our results support the hypothesis that adolescents with GD experience a “clinically significant distress” (*DSM 5*) [2], due to the mismatch between the individual’s experienced and/or expressed gender and the birth assigned gender. So, this study emphasizes the intrinsic emotional suffering linked with the awareness of gender incongruity, which is compounded by environmental challenges such as marginalization, discrimination, rejection, violence, and transphobia. Both of these components play a role in the development and maintenance of internalizing disorders, self-harm, and suicide risk, and are exacerbated by the problems that come with adolescent challenges. The care of GD adolescents must take into account these factors, without neglecting the function of emotion regulation and affective investment in the body in the genesis and maintenance of internalizing psychopathological disorders. Acting on the aforementioned predictive criteria would make the management more detailed, more customized, and definitely more effective. Future research should be presented with longitudinal designs and should use more generally recognized and standardized measurements.

## Figures and Tables

**Table 1 jcm-11-03314-t001:** Clinical features for GD sample.

Gender Dysphoria Clinical Sample	
**Age of the SR request**	
(mean ± S.D.)	15.6 ± 1.6
**Gender**	
Male-assigned at birth (MtoF) *n* (%)	7 (23.33%)
Female-assigned at birth (FtoM) *n* (%)	23(76.67%)
**Neurodevelopmental disorders history***n* (%)	7 (23.33%)
**Bullying suffered***n* (%)	7 (23.33%)
**GD onset**	
Childhood onset *n* (%)	17 (56.67%)
Adolescent onset *n* (%)	13 (43.33%)
**NSSI and Suicidal behavior**	
NSSI behavior *n* (%)	14 (46.67%)
Suicidal ideation *n* (%)	19 (63.33%)
Suicidal acting *n* (%)	4 (13.33%)
**Psychopathological symptoms history**	
Anxiety *n* (%)	22 (73.33%)
Depression *n* (%)	27 (90%)
Socially withdrawn *n* (%)	21 (70%)
Substance abuse *n* (%)	3 (10%)
Eating disorders *n* (%)	11 (36.67%)
Sleep disorders *n* (%)	16 (53.33%)
Attention deficit *n* (%)	18 (60%)
**Previous psychotherapy***n* (%)	16 (53.33%)

**Table 2 jcm-11-03314-t002:** Mann–Whitney comparison of SAFA, BIS, and DERS scales between GD sample and cisgender volunteers.

	GD Sample *n* = 30		Cisgender Volunteers *n* = 30			
	m	Rank-Av	m	Rank-Av	U	*p* Values
** *SAFA* **						
*SAFA-A*	55.77	19.60	34.40	41.40	777.000	**<0.001**
*SAFA-D*	66.50	18.72	48.60	42.28	803.500	**<0.001**
*SAFA-O*	53.30	29.02	50.50	31.98	494.500	0.510
*SAFA-P*	55.43	27.82	51.50	33.18	530.500	0.232
*SAFA-S*	58.90	20.43	46.97	40.57	752.000	**<0.001**
** *BIS* **						
*BIS-C*	17.03	25.14	20.10	24.33	265.000	**0.006**
*BIS-T*	17.80	24.92	19.77	28.10	378.000	0.285
*BIS-P*	19.83	26.50	24.57	22.52	210.500	**<0.001**
*BIS-I*	14.43	19.06	26.20	16.98	44.500	**<0.001**
** *DERS* **						
*DERS-TOT*	106.37	22.17	78.67	38.83	700.000	**<0.001**
*D-Non-Accept*	15.20	26.02	11.47	34.98	584.500	**0.046**
*D-Goals*	19.43	23.37	14.80	37.63	664.000	**0.002**
*D-Impulse*	15.80	27.97	13.50	33.03	526.000	0.259
*D-Awareness*	16.90	26.27	13.67	34.73	577.000	0.06
*D-Strategies*	24.03	21.38	15.97	39.62	723.500	**<0.001**
*D-Clarity*	15.03	20.83	9.23	40.17	740.000	**<0.001**

Values are shown as means and rank-average. Bold font is indicative of *p* < 0.05.

**Table 3 jcm-11-03314-t003:** Spearman’s correlations between DERS total and subscales and BIS domains among GD sample.

		BIS-C	BIS-P	BIS-T	BIS-I
*D-TOT*	r_s_ Sig. (2-tails)	−0.281 0.132	**−0.530 **** **0.003**	−0.141 0.459	−0.118 0.533
*D-Non-Accept*	r_s_ Sig. (2-tails)	−0.057 0.765	−0.350 0.058	−0.174 0.357	−0.027 0.886
*D-Goals*	r_s_ Sig. (2-tails)	−0.201 0.286	−0.113 0.553	0.015 0.936	0.000 0.999
*D-Impulse*	r_s_ Sig. (2-tails)	0.096 0.616	−0.285 0.126	0.207 0.273	−0.057 0.764
*D-Awareness*	r_s_ Sig. (2-tails)	−0.352 0.056	**−0.389 *** **0.034**	−0.358 0.052	−0.302 0.104
*D-Strategy*	r_s_ Sig. (2-tails)	−0.199 0.292	**−0.510 **** **0.004**	−0.139 0.465	0.000 1.000
*D-Clarity*	r_s_ Sig. (2-tails)	**−0.545 **** **0.002**	**−0.498 **** **0.005**	−0.210 0.265	−0.250 0.182

Bold fonts are indicative of significance. * corresponds to *p* < 0.05, ** correspond to *p* < 0.01.

**Table 4 jcm-11-03314-t004:** Univariate logistic binary regression’s results for each SAFA scale as dependent variable and DERS-TOT and BIS domains as independent variables: **A**. gender dysphoria sample; **B**. cisgender volunteers.

A								
*GD Sample*								
*SAFA-A (cut-off > 60: n = 13)*								
	*B*	*S.E.*	*Wald*	*gl*	*Sign.*	*Exp(B)*	*C.I. Inf.*	*C.I. Sup*
** *DERS-TOT* **	0.036	0.017	4.523	1	**0.033**	1.037	1.003	1.072
*Constant*	−4.214	1.933	4.753	1	**0.029**	0.015		
** *BIS-C* **	−0.056	0.094	0.352	1	0.553	0.946	0.787	1.137
*Constant*	0.677	1.631	0.172	1	0.678	1.968		
** *BIS-T* **	−0.002	0.066	0.066	1	0.066	0.998	0.877	1.136
*Constant*	−0.237	1.228	0.037	1	0.847	0.789		
** *BIS-P* **	−0.124	0.082	2.322	1	0.128	0.883	0.753	1.036
*Constant*	2.188	1.661	1.735	1	0.188	8.92		
** *BIS-I* **	0.022	0.096	0.051	1	0.821	1.022	0.847	1.232
*Constant*	−0.580	1.431	0.164	1	0.685	0.56		
** *SAFA-D (cut-off > 60: n = 21)* **								
	*B*	*S.E.*	*Wald*	*gl*	*Sign.*	*Exp(B)*	*C.I. Inf.*	*C.I. Sup*
** *DERS-TOT* **	0.057	0.022	6.891	1	**0.009**	1.059	1.015	1.105
*Constant*	−4.855	2.128	5.203	1	**0.023**	0.008		
** *BIS-C* **	−0.128	0.112	1.32	1	0.251	0.88	0.707	1.095
*Constant*	3.086	2.034	2.303	1	0.129	21.895		
** *BIS-T* **	−0.009	0.071	0.016	1	0.898	0.991	0.862	1.14
*Constant*	1.011	1.339	0.57	1	0.45	2.747		
** *BIS-P* **	−0.739	0.278	7.07	1	**0.008**	0.478	0.277	0.823
*Constant*	17.077	6.336	7.265	1	**0.007**	2.610 × 10^7^		
** *BIS-I* **	−0.054	0.104	0.275	1	0.6	0.947	0.773	1.161
*Constant*	1.641	1.583	1.075	1	0.3	5.162		
** *SAFA-O (cut-off > 60: n = 9)* **								
	*B*	*S.E.*	*Wald*	*gl*	*Sign.*	*Exp(B)*	*C.I. Inf.*	*C.I. Sup*
** *DERS-TOT* **	0.051	0.023	5.104	1	**0.024**	1.052	1.007	1.1
*Constant*	−6.635	2.718	5.96	1	**0.015**	0.001		
** *BIS-C* **	0.017	0.101	0.029	1	0.865	1.017	0.835	1.24
*Constant*	−1.142	1.777	0.413	1	0.521	0.319		
** *BIS-T* **	0.067	0.074	0.812	1	0.368	1.069	0.924	1.236
*Constant*	−2.065	1.44	2.058	1	0.151	0.127		
** *BIS-P* **	−0.046	0.078	0.348	1	0.555	0.955	0.821	1.112
*Constant*	0.05	1.558	0.001	1	0.975	1.051		
** *BIS-I* **	0.11	0.107	1.052	1	0.305	1.116	0.905	1.376
*Constant*	−2.466	1.661	2.205	1	0.138	0.085		
** *SAFA-P (cut-off > 60: n = 12)* **								
	*B*	*S.E.*	*Wald*	*gl*	*Sign.*	*Exp(B)*	*C.I. Inf.*	*C.I. Sup*
** *DERS-TOT* **	0.067	0.026	6.898	1	**0.009**	1.069	1.017	1.124
*Constant*	−7.865	2.972	7.004	1	**0.008**	0		
** *BIS-C* **	−0.174	0.106	2.704	1	0.1	0.84	0.682	1.034
*Constant*	2.529	1.813	1.946	1	0.163	12.538		
** *BIS-T* **	−0.057	0.068	0.693	1	0.405	0.945	0.827	1.08
*Constant*	0.594	1.248	0.227	1	0.634	1.812		
** *BIS-P* **	−0.223	0.101	4.869	1	**0.027**	0.8	0.656	0.975
*Constant*	3.982	2.044	3.795	1	**0.051**	53.64		
** *BIS-I* **	−0.011	0.097	0.013	1	0.908	0.989	0.818	1.195
*Constant*	−0.244	1.44	0.029	1	0.865	0.783		
** *SAFA-S (cut-off > 60: n = 12)* **								
	*B*	*S.E.*	*Wald*	*gl*	*Sign.*	*Exp(B)*	*C.I. Inf.*	*C.I. Sup*
** *DERS-TOT* **	0.046	0.019	5.492	1	**0.019**	1.047	1.007	1.087
*Constant*	−5.407	2.235	5.852	1	**0.016**	0.004		
** *BIS-C* **	0.041	0.095	0.185	1	0.667	1.042	0.864	1.256
*Constant*	−1.107	1.679	0.435	1	0.51	0.331		
** *BIS-T* **	0.024	0.067	0.129	1	0.719	1.024	0.898	1.168
*Constant*	−0.836	1.259	0.44	1	0.507	0.434		
** *BIS-P* **	−0.101	0.078	1.665	1	0.197	0.904	0.776	1.054
*Constant*	1.578	1.579	0.998	1	0.318	4.843		
** *BIS-I* **	0.073	0.098	0.558	1	0.455	1.076	0.888	1.305
*Constant*	−1.473	1.487	0.981	1	0.322	0.229		
**B**								
** *Cisgender Volunteers* **								
** *SAFA-A (cut-off > 60: n = 3)* **								
	*B*	*S.E.*	*Wald*	*gl*	*Sign.*	*Exp(B)*	*C.I. Inf.*	*C.I. Sup*
** *DERS-TOT* **	0.04	0.03	1.74	1	0.187	1.04	0.981	1.104
*Constant*	−5.560	2.799	3.945	1	0.047	0.004		
** *BIS-C* **	−0.114	0.165	0.476	1	0.49	0.892	0.646	1.233
*Constant*	0.018	3.167	0	1	0.995	1.018		
** *BIS-T* **	0.035	0.143	0.06	1	0.807	1.035	0.783	1.369
*Constant*	−2.895	2.954	0.96	1	0.327	0.055		
** *BIS-P* **	−0.248	0.174	2.03	1	0.154	0.781	0.555	1.098
*Constant*	3.534	3.848	0.843	1	0.358	34.276		
** *BIS-I* **	0.162	0.152	1.132	1	0.287	1.176	0.873	1.584
*Constant*	−6.688	4.461	2.248	1	0.134	0.001		
** *SAFA-D (cut-off > 60: n = 2)* **								
	*B*	*S.E.*	*Wald*	*gl*	*Sign.*	*Exp(B)*	*C.I. Inf.*	*C.I. Sup*
** *DERS-TOT* **	0.028	0.035	0.651	1	0.42	1.028	0.961	1.101
*Constant*	−4.980	3.161	2.482	1	0.115	0.007		
** *BIS-C* **	−0.758	0.471	2.593	1	0.107	0.469	0.186	1.179
*Constant*	10.173	7.342	1.92	1	0.166	26,191.43		
** *BIS-T* **	0.014	0.172	0.007	1	0.936	1.014	0.724	1.421
*Constant*	−2.915	3.518	0.687	1	0.407	0.054		
** *BIS-P* **	−1.242	0.861	2.081	1	0.149	0.289	0.053	1.561
*Constant*	22.328	16.161	1.909	1	0.167	4.975 × 10^9^		
** *BIS-I* **	−0.063	0.133	0.224	1	0.636	0.939	0.723	1.219
*Constant*	−1.035	3.365	0.095	1	0.758	0.355		
** *SAFA-O (cut-off > 60: n = 3)* **								
	*B*	*S.E.*	*Wald*	*gl*	*Sign.*	*Exp(B)*	*C.I. Inf.*	*C.I. Sup*
** *DERS-TOT* **	0.162	0.088	3.367	1	0.067	1.176	0.989	1.398
*Constant*	−17.728	9.109	3.787	1	0.052	0		
** *BIS-C* **	−0.034	0.162	0.045	1	0.832	0.966	0.703	1.327
*Constant*	−1.514	3.252	0.217	1	0.642	0.22		
** *BIS-T* **	−0.048	0.146	0.109	1	0.741	0.953	0.715	1.269
*Constant*	−1.258	2.858	0.194	1	0.66	0.284		
** *BIS-P* **	−0.314	0.19	2.722	1	0.099	0.731	0.503	1.061
*Constant*	4.953	4.099	1.46	1	0.227	141.627		
** *BIS-I* **	−0.021	0.114	0.035	1	0.852	0.979	0.782	1.225
*Constant*	−1.643	3.005	0.299	1	0.585	0.193		
** *SAFA-P (cut-off > 60: n = 2)* **								
	*B*	*S.E.*	*Wald*	*gl*	*Sign.*	*Exp(B)*	*C.I. Inf.*	*C.I. Sup*
** *DERS-TOT* **	0.062	0.039	2.485	1	0.115	1.064	0.985	1.15
*Constant*	−8.191	3.987	4.221	1	0.04	0		
** *BIS-C* **	−0.758	0.471	2.593	1	0.107	0.469	0.186	1.179
*Constant*	10.173	7.342	1.92	1	0.166	26,191.43		
** *BIS-T* **	0.014	0.172	0.007	1	0.936	1.014	0.724	1.421
*Constant*	−2.915	3.518	0.687	1	0.407	0.054		
** *BIS-P* **	−0.818	0.505	2.617	1	0.106	0.441	0.164	1.189
*Constant*	14.34	9.627	2.219	1	0.136	########		
** *BIS-I* **	−0.134	0.134	0.995	1	0.319	0.875	0.673	1.138
*Constant*	0.642	3.158	0.041	1	0.839	1.9		
** *SAFA-S (cut-off > 60: n = 2)* **								
	*B*	*S.E.*	*Wald*	*gl*	*Sign.*	*Exp(B)*	*C.I. Inf.*	*C.I. Sup*
** *DERS-TOT* **	0.028	0.035	0.651	1	0.42	1.028	0.961	1.101
*Constant*	−4.980	3.161	2.482	1	0.115	0.007		
** *BIS-C* **	−0.758	0.471	2.593	1	0.107	0.469	0.186	1.179
*Constant*	10.173	7.342	1.92	1	0.166	26,191.43		
** *BIS-T* **	0.014	0.172	0.007	1	0.936	1.014	0.724	1.421
*Constant*	−2.915	3.518	0.687	1	0.407	0.054		
** *BIS-P* **	−1.242	0.861	2.081	1	0.149	0.289	0.053	1.561
*Constant*	22.328	16.161	1.909	1	0.167	4.975 × 10^9^		
** *BIS-I* **	−0.063	0.133	0.224	1	0.636	0.939	0.723	1.219
*Constant*	−1.035	3.365	0.095	1	0.758	0.355		

## Data Availability

The data presented in this study are available on request from the corresponding author.

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
