# Peer review of "Body Emotional Investment and Emotion Dysregulation in a Sample of Adolescents with Gender Dysphoria Seeking Sex Reassignment"

_jcm, 2022, doi:10.3390/jcm11123314_

Round 1

Reviewer 1 Report

Thank you for asking me to review this paper.

The objectives of this study were: 1) to compare the Self-Administrated Psychiatric Scales for Children and Adolescents' (SAFA), Body Investment Scale's (BIS) and Difficulties in Emotion Regulation Scale's (DERS) scores between GD Adolescents (N=30) and Cisgenders (N=30), 2) to find correlations between body investment and emotion regulation in the GD sample, 3) to evaluate the predictive role of these dimensions on internalizing symptomatology of GD adolescents.

The authors showed altered scores in adolescent GD compared to controls, an association between worse body protection and impaired emotion regulation, and an association between these 2 dimensions and depression.

The subject is interesting and the article well written. However, I have few comments regarding the study:

Recruitment:

1. Do the GD adolescents included correspond to all the people likely to be included during the 24-month inclusion period? If not, what proportion did not answer, and what are the reasons for exclusion?

2. Where and how were volunteers selected?

Models:

Regarding the model, I have several questions:

3. In the methods section, it is written “Moreover, a stepwise logistic binary regression was performed to estimate the contribution of each BIS domains and DERS-TOT as predictors and each SAFA scale as dependent variable using T=60 as cut-off for pathological values.”

But in Table 4, it is written “Results accepted in the model of Logistic Binary Regression for each SAFA scale as independent variable and DERS-TOT and BIS-P as dependent variables.”

If I understood correctly, there is an error in Table 4 title. I understand that the authors carried out one logistic regression for each SAFA subscale (5 models), considered as the dependent variable, and tried to explain SAFA subscale with DERS and BIS as predictors: SAFA (subscale X) ~ SAFA subscales + BIS subscales.

4. The selection of variables using a stepwise regression is highly unstable and not recommended. especially in a small sample size. Could the authors present all the results for each of the predictors?

5. Why did the authors choose to perform logistic regressions with a threshold for the SAFA scales?

Furthermore, it seems that the number of events will not allow to verify the conditions for carrying out logistic regressions (5-10 events per variable included in the model).

6. If the authors justify this choice and maintain it: how is the threshold of 60 justified? Can the authors provide the number of participants with a score greater than 60 for each of the SAFA subscales?

7. Why not use linear regressions which won't cause sample size issues and will limit power loss? Maybe the authors preferred to perform a logistic regression because the distributions of the SAFA subscales were not normal. However, if the errors of a linear regression model are normally distributed, we can suppose that the “normality assumption” is fulfilled for linear regression. Pending the authors' answers, it seems that, in this case, linear regression models would be more appropriate than logistic regression models.

8. Has the collinearity between the explanatory variables been verified, with calculations of variance inflation factors (VIFs) for example?1

 1Johnston R, Jones K, Manley D. Confounding and collinearity in regression analysis: a cautionary tale and an alternative procedure, illustrated by studies of British voting behaviour. Qual Quant. 2018;52(4):1957–76.

9. Are the results found in the GD group similar to those of the control group? The association identified between SAFA and BIS/DERS could be similar in both groups.

It seems difficult to carry out the analyzes on all adolescents (GD and control), including a "group" variable and defining interactions between predictors and group. However, the analyzes could be stratified on the group and table 4 could present the results for both GD adolescents and controls.

Discussion:

10. The lack of collected data that does not allow potential confounding factors to be considered should also be mentioned in the limitations section.

11. In addition, a precaution on the predictive aspect of the variables must be added, due to the cross-sectional design of the study.

Author Response

Dear Revisor, thank you for the useful suggestions and questions about our paper. We are giving our explanations point by point and together with adjustments in the manuscript according with your comments:

Recruitment:

  1. Do the GD adolescents included correspond to all the people likely to be included during the 24-month inclusion period? If not, what proportion did not answer, and what are the reasons for exclusion?

We are thankful for this advice; we modified the manuscript in the result section according with the following answer to your question: the total number of adolescents who came to our attention with a request of sex reassignment during the 24-month inclusion period is of 33 units. Two of them were excluded because they had a diagnosis of autism spectrum disorder, which is an exclusion criteria for our study; moreover, one adolescent dropped out prior to the underwent to psychometric assessment.

  1. Where and how were volunteers selected?

Actually, explanations about comparison group recruitment were missing. We added in the methods section a brief description explaining that the subjects of the comparison group were recruited in two territorial High Schools. Adolescents without psychopathological history were asked to complete the assessment questionnaires as volunteers after we presented the project and after the obtainment of written informed consent from parents and adolescents themselves.

Models:

Regarding the model, I have several questions:

  1. In the methods section, it is written “Moreover, a stepwise logistic binary regression was performed to estimate the contribution of each BIS domains and DERS-TOT as predictors and each SAFA scale as dependent variable using T=60 as cut-off for pathological values.”

But in Table 4, it is written “Results accepted in the model of Logistic Binary Regression for each SAFA scale as independent variable and DERS-TOT and BIS-P as dependent variables.”

If I understood correctly, there is an error in Table 4 title. I understand that the authors carried out one logistic regression for each SAFA subscale (5 models), considered as the dependent variable, and tried to explain SAFA subscale with DERS and BIS as predictors: SAFA (subscale X) ~ SAFA subscales + BIS subscales.

We are really sorry for the mistake. As you said it was performed a logistic regression for each SAFA subscale (5 models), considered as the dependent variable, and tried to explain SAFA subscale with DERS and BIS as predictors. We are providing a revised form of the manuscript in which the caption of Table 4. is modified according to corrections about this point.

  1. The selection of variables using a stepwise regression is highly unstable and not recommended. especially in a small sample size. Could the authors present all the results for each of the predictors?

We would like to thank the reviewer for the appropriate comment that allowed us to further reflect on the opportunity to perform the selection of variables using stepwise regression, in particular for the risks of our sample size. Starting from your correct observation, we tried to improve the stability of our model, according to the specific aims of this study, replacing the stepwise regression with univariate binary logistic regressions, in order to evaluate the predictive capacity of each independent variable separately towards the SAFA subscales. As you requested, we are presenting all the results for each of the predictors in an improved version of Table 4.

As you will see in the new version of the Table 4 the results obtained from univariate logistic regressions are similar to the stepwise method, but of course they are obtained from a more stable statistical procedure that is less influenced from the small sample size than stepwise method in which the criteria of 5-10 events per variable included in the model was not respected.

The only difference is about the statistically significant relationship between a worse attitude in the body protection (BIS-P) and the presence of eating disorders (SAFA-P), even if this relationship seems to be weaker than with the presence of depression (SAFA-D). Anyway, this new finding is confirmed by clinical observations and is in the same line with previous obtained results, so, it is commented too in the new version of the manuscript.

Below we give you a combined response to the points 5 and 6, since they refer to the same focus:

  1. Why did the authors choose to perform logistic regressions with a threshold for the SAFA scales?

Furthermore, it seems that the number of events will not allow to verify the conditions for carrying out logistic regressions (5-10 events per variable included in the model).

  1. If the authors justify this choice and maintain it: how is the threshold of 60 justified? Can the authors provide the number of participants with a score greater than 60 for each of the SAFA subscales?

We’ll try to better explain our choice to perform logistic regression with a threshold for the SAFA scales.

SAFA is an Italian battery of self-administered psychiatric scales for children and adolescents exploring symptoms of anxiety-related areas, depression-related areas, obsessive-compulsive areas, somatic concerns, and psychogenic eating disorders, and its answers were evaluated by transforming the raw scores into standardized T scores. According to the structure and validation of the SAFA questionnaire (See: Franzoni E, Monti M, Pellicciari A, Muratore C, Verrotti A, Garone C, Cecconi I, Iero L, Gualandi S, Savarino F, Gualandi P. SAFA: A new measure to evaluate psychiatric symptoms detected in a sample of children and adolescents affected by eating disorders. Correlations with risk factors. Neuropsychiatr Dis Treat. 2009;5:207-14. doi: 10.2147/ndt.s4874. Epub 2009 Apr 8. PMID: 19557115; PMCID: PMC2695231. Or Pellicciari A, Gualandi S, Iero L, Monti M, Di Pietro E, Sacrato L, Gualandi P, Franzoni E. Psychometric evaluation of SAFA P test for eating disorders in adolescents: comparative validation with EDI-2. Eur Eat Disord Rev. 2012 Jan;20(1):e108-13. doi: 10.1002/erv.1099. Epub 2011 Feb 9. PMID: 21308872), T scores >60 were assumed as the cut-off used to consider if the domain of interest was pathological. Thus, we choose to perform logistic regressions to verify whether the scores obtained on DERS-TOT and each BIS subscales were significantly related to the presence/absence of symptoms of pathological relevance in the areas of anxiety, depression, obsessions, eating-related behaviors, and somatic problems, through the dichotomization of SAFA with the cut-off of 60.

As you request, we added the number of participants with a score greater than 60 for each of the SAFA subscales in Table 4.

  1. Why not use linear regressions which won't cause sample size issues and will limit power loss? Maybe the authors preferred to perform a logistic regression because the distributions of the SAFA subscales were not normal. However, if the errors of a linear regression model are normally distributed, we can suppose that the “normality assumption” is fulfilled for linear regression. Pending the authors' answers, it seems that, in this case, linear regression models would be more appropriate than logistic regression models.

About the opportunity to choose the linear regression, we maintained the choice to operate a binary logistic regression for the reasons set out above. In addition, in consideration of the limits of stepwise method we have decided to make the statistics more reliable and powerful by adopting a univariate analysis method for each independent variable, which has the advantage of being less affected by the narrowness of the sample size. Moreover, thanks to your suggestions, we had the opportunity to examine more carefully the hypotheses and how to test them and we recognized that the univariate model is better suited to the purposes of our study, i.e., to examine the individual predictive capacity of each of the domains of the BIS and DERS TOT scales towards the SAFA subscales.

  1. Has the collinearity between the explanatory variables been verified, with calculations of variance inflation factors (VIFs) for example?1

 1Johnston R, Jones K, Manley D. Confounding and collinearity in regression analysis: a cautionary tale and an alternative procedure, illustrated by studies of British voting behaviour. Qual Quant. 2018;52(4):1957–76.

Thank you for this suggestion and reference about the collinearity. We tested it and each variable has a VIF really close to 1, Thank you for this advice, we are putting the assumption of non-collinearity in the method paragraph of our manuscript.

  1. Are the results found in the GD group similar to those of the control group? The association identified between SAFA and BIS/DERS could be similar in both groups.

It seems difficult to carry out the analyzes on all adolescents (GD and control), including a "group" variable and defining interactions between predictors and group. However, the analyzes could be stratified on the group and table 4 could present the results for both GD adolescents and controls.

we followed your advice to verify the association between SAFA and BIS/DERS also in the control group. The results are shown in the extended form of Table.4. As you can see, no one of them have a p-value<0,05.

Discussion:

  1. The lack of collected data that does not allow potential confounding factors to be considered should also be mentioned in the limitations section.

According to your suggestion We are upgrading and revising the entire paragraph of limitations, also adding this specific point. Unfortunately, the retrospective way of data collection doesn’t allow to analyze potential clinical and anamnestic confounding factors.

  1. In addition, a precaution on the predictive aspect of the variables must be added, due to the cross-sectional design of the study.

We are adding another point in the paragraph of limitations regarding the risks to perform causal interpretations on the relationship between psychopathological dimensions examined by SAFA and BIS and DERS as predictors. Moreover, we are revising the manuscript to make some corrections in the lines in which this association is too explicit or written in the form of causal interpretation. 

Reviewer 2 Report

The authors report an interesting study on body investment in adolescents (12-18) with gender dysphoria (measured at  a specialized center for gender dysphoria in Bari (cfr. line 91/92 of the manuscript). Compared at 1 timepoint were a group of 30 "GD adolescents" ["23 natal females and 7 natal males"] with 30 "healthy Cisgenders" [26 girls and 4 boys] using the Self-Administered Psychiatric Scale for Children and Adolescents , the Body Investment Scale and Difficulties in Emotion Regulation Scale. The authors concluded that the adolescents with GD had "significant impairment of both emotion regulation and emotional investment in the body, along with the presence of internalizing symptoms of anxiety, depression, and somatization".

Although the manuscript is interesting and the reported research has some innovative aspects (as outlined by the authors), and the used statistics are sound,  the manuscript has important shortcomings. First we will outline major shortcomings; followed by some minor problems.

Major shortcomings

(1) The sample size is problematic. Not only the size of the total sample, but also of the "subgroups". Only 7 "natal females" and 4 "boys" were included in the two used comparison groups of 30. Why didn't the authors collect enough participants to fullfill power requirements?   In the view of this reviewer it is better to have a sufficient number of participants to fulfill power requirements than to use inadequate samples. And why did the authors  included  only 7 "natal males" and "4 boys".  Furthermore, given the fact that only 7 "natal males" and  4 "boys" were included, it would have been better to restrict the comparison to "natal females" and "girls". And in consequence, to exclude the boys and "natal males" from the analyses and comparisons between the two groups. 

(2) Given the fact that the authors used only 1 measurement time-point, the  design does not allow causal interpretations, which unfortunately the authors make more than once (see minor problems for examples)

(3) The authors use DSM 5 not always correctly. They write: "Gender dysphoria" (GD) is defined as a subjective experience of marked discrepancy between the experienced gender and individual sex". (line 37/38). However, that is not what DSM 5 (2013) specifies on page 451: "Gender dysphoria as a general descriptive term refers to an individual's affective/cognitive discontent with the assigned gender....Gender dysphoria refers to the distress that may accompany the incongruence between one's experienced or expressed gender and one's assigned gender". (See also the Diagnostic Criteria for Gender Dysphoria on page 452: They do not refer to an incongruence with an individual's sex; but refer only to primary and/or secondary sex characteristics (or in young adolescents, the anticipated secondary sex characteristics). If the authors would like to use DSM 5; please use it correct and change the manuscript accordingly.

(4) The authors use some undefined terms: 

They use for example "natal female" and "natal male". Besides the fact that not everybody will agree that this is the best terminology (see for example the work of T'Sjoen and Bouman on terminology), the authors do not define how they define "a natal male" and "a natal female". And why are the authors using this term "natal"? The point here is not that the authors should not use this term, but that they should be  explicit why these use certain terms and define them transparantly/explicitly. Another example is the use of the undefined "outward sex". 

Along the same line: why do the authors write at the beginning: "Gender identity refers to a person's internal feelings about gender, such as boy or girl, man or woman. The term "Gender variance". Why do the authors restrict their examples of gender identity in the first sentence to the categories "boy, girl, man or woman"? 

(5)Although the authors report Cronbach alpha's for the used measures, it is not clear in their manuscript to what extent the used scales have been validated in Italy. (see further minor points)

(6)The authors use as a comparison group: "cisgender healthy volunteers", but do not specifiy what the word healthy means. Does it refer to "the absence of a somatic disease (in the traditional meaning of the word) or does it refer to "no psychopathology".

(7)How were the cisgender healthy volunteers recruited. Please specify.

Minor problems:

(1)Line 48: The authors wrote that the numbers of referrals to specialized gender clinics has increased. Just writing increased is very vague and not much informative. Please specify very shortly this trend in numbers (in general or for your own center). There have been papers published on the general trends. To give but one example: See Wiepjes et al. (2018). The Amsterdam cohort of gender dysphoria study (1972-2015). Journal of Sexual Medicine, 15: 582-590.

(2) Line 55: "at rates higher". Please specify how much higher in numbers. At rates higher remains very vague. And specify if this is the case for only untreated adolescents with GD or also the case for treated adolescents (Compare for example with: Arnoldussen, M. et al. (2022). Self-perception of transgender adolescents after gender-affirming treatment: A follow-up into young adulthood. LGBT Health).

(3)Line 59: "we argued that that". Should that not be: We argue that.

(4)Line 66: What do you mean with the construct "tendency to be attracted to life"?

(5)Line 83: "with a control group of healthy cisgender volunteers". Please replace control by comparison group. And  do so in the whole manuscript. 

(6)Line 93: "over a period of 24 months". Please specifiy the date when this period started, respectively ended.

(7)Line 96 and 97: Why were Intellectual Disability and Autism Spectrum Disorder used as exclusion cirteria. Please specify. And why were other psychiatric syndromes (as for example bordeline personality disorder, or gilles de la tourette not used as exlusion criteria). Furthermore, please discuss in the general discussion the repercussions of the used exclusion criteria on the (lack of ?) representativeness of the sample.

(8)Line 100/101: Please specify in a concrete way what you mean by neurodevelopmental disorders and how they were differentiated from psychiatric symptoms.

(9)Line 105 and 107: Psychopathological assessment. The reviewer suggest to use: Assessment of psychopathology instead of Psychopathological assessment.

(10)Line 109: SAFA: Is this an Italian scale or a scale translated in Italian? Are the reported Cronbach's alpha calculated on the original version or on the Italian data? Besides Chronbachs alpha, have there been studies validating this scale in other ways in the Italian version?

(11)For the other used scales the same question: Have there been studies validating the scales in other ways (for example divergent, convergent or predictive validity) using Italian data? 

(12)Line 190-193: "both have a role" . What do the authors mean by both have a role. It seems they are suggesting causality, based on 1 measurement point. But, one measurement does not allow such causal interpretation. The same holds for the causal interpretations in the next paragraph. Furthermore, it seems that the authors are of the opinion that what applies for body image dissatisfaction applies also for body investment. Why, given that both constructs are not the same?

(13)Line: 207: What do the authors mean by maturation of body incongruence awareness? For example, why didn't they use the intensification of ...?

(14)Line 235-238: The limitations of the study are discussed in a very superficial way. Please elaborate the limitations. To give one example: to what extent are the results influenced by the specific cultural context in which the research was done? To give another example: what are the limitations of the used in- and exclusion criteria? See also earlier made comments. And, specify also the strenghts of the study!

Author Response

Dear Revisor, I would like to thank you for the useful suggestions and questions about our paper. We are giving our explanations point by point and together with adjustments in the manuscript according with major and minor problems you underlined.

Major shortcomings

(1) The sample size is problematic. Not only the size of the total sample, but also of the "subgroups". Only 7 "natal females" and 4 "boys" were included in the two used comparison groups of 30. Why didn't the authors collect enough participants to fullfill power requirements?   In the view of this reviewer it is better to have a sufficient number of participants to fulfill power requirements than to use inadequate samples. And why did the authors included only 7 "natal males" and "4 boys".  Furthermore, given the fact that only 7 "natal males" and  4 "boys" were included, it would have been better to restrict the comparison to "natal females" and "girls". And in consequence, to exclude the boys and "natal males" from the analyses and comparisons between the two groups. 

We are really thankful about the pertinent considerations about this topic. As you said, we are aware that the small sample size is risky about the power lost, but we know that Gender Dysphoria is a rare condition, and even more important is the exiguity of adolescents who came to special services with a clear request of sex reassignment, making difficult to collect large samples. Moreover, in the last period the recruitment proceeded very slow also for Covid-19 pandemics. Even if we are aware of the limitations related to the small sample size, we hope that, according to the increasing interest on this matter associated with the exiguity of the literature on this topic, you would take in consideration our paper for eventual publication. Regarding the "subgroups", we decided to keep in the recruitment also the few subjects assigned-male at birth, to avoid further reduction of the sample. On the other hand the sex-ratio we have in our small group reflects the epidemiological data in adolescence and corresponds to the naturalistic description of the sex reassignment requests received by our services. In our opinion, for the purpose of our study we think that this doesn’t affect significantly the quality of the results. Moreover, considering appropriate your notice we added in the results section the comparison about the sex proportion between the two groups (p=0,317) to make clearer that the sex ratio was homogeneous among study and control groups.

(2) Given the fact that the authors used only 1 measurement time-point, the design does not allow causalinterpretations, which unfortunately the authors make more than once (see minor problems for examples)

We are sorry we have been too explicit in the use of causative inferences even if study architecture doesn’t allow it. As a result we revised the manuscript to make some corrections in the lines in which this association is too explicit or written in the form of causal interpretation. Moreover, we added a clarification in the paragraph of limitations regarding the risks to perform causal interpretations on the relationship between psychopathological dimensions examined by SAFA and BIS and DERS-tot as predictors due to the cross-sectional nature of the study.

(3) The authors use DSM 5 not always correctly. They write: "Gender dysphoria" (GD) is defined as a subjective experience of marked discrepancy between the experienced gender and individual sex". (line 37/38). However, that is not what DSM 5 (2013) specifies on page 451: "Gender dysphoria as a general descriptive term refers to an individual's affective/cognitive discontent with the assigned gender....Gender dysphoria refers to the distress that may accompany the incongruence between one's experienced or expressed gender and one's assigned gender". (See also the Diagnostic Criteria for Gender Dysphoria on page 452: They do not refer to an incongruence with an individual's sex; but refer only to primary and/or secondary sex characteristics (or in young adolescents, the anticipated secondary sex characteristics). If the authors would like to use DSM 5; please use it correct and change the manuscript accordingly.

We revised the manuscript to be more adherent to DSM-5 as you suggested. So, we changed the LINES 37-38 with the following sentence:Gender dysphoria is a general descriptive term that refers to an individual’s affective/cognitive discontent with the assigned gender; meanwhile in a diagnostic manner it refers to the distress that may accompany the incongruence between one’s experienced or expressed gender and one’s assigned gender”.

(4) The authors use some undefined terms: 

They use for example "natal female" and "natal male". Besides the fact that not everybody will agree that this is the best terminology (see for example the work of T'Sjoen and Bouman on terminology), the authors do not define how they define "a natal male" and "a natal female". And why are the authors using this term "natal"? The point here is not that the authors should not use this term, but that they should be  explicit why these use certain terms and define them transparantly/explicitly. Another example is the use of the undefined "outward sex". 

Along the same line: why do the authors write at the beginning: "Gender identity refers to a person's internal feelings about gender, such as boy or girl, man or woman. The term "Gender variance". Why do the authors restrict their examples of gender identity in the first sentence to the categories "boy, girl, man or woman"? 

Reading the work that you mentioned, it has been very helpful to understand clearly all the aspects of terminology. So, we modified the manuscript according to your suggestion preferring to be more explicit: we are changing “female-assigned at birth” and “male-assigned at birth” instead of “natal female” and “natal male”. About the term “outward sex” we are upgrading the entire sentence as the following: “The term “Gender Variance” includes a wide spectrum of gender experiences and behaviors pointing to a partial or complete mismatch between an individual's gender identity and the sex established at birth”.

Moreover, we revised the beginning sentence to be more adherent with DSM-5: …Gender identity refers to an individual’s identification as male, female, or, occasionally, some category other than male or female (pag. 451)…

(5) Although the authors report Cronbach alpha's for the used measures, it is not clear in their manuscript to what extent the used scales have been validated in Italy. (see further minor points)

We are sorry that this information is not so clear and explicit in the manuscript; we revised the section of assessment in order to make explicit this clarification.

(6)The authors use as a comparison group: "cisgender healthy volunteers", but do not specifiy what the word healthy means. Does it refer to "the absence of a somatic disease (in the traditional meaning of the word) or does it refer to "no psychopathology".

We are thankful for the suggestion to better specify the correct qualification of the comparison group. We replaced the term “healthy” with the description of “with no clear previous or current history of formal psychopathological disorders”.

(7) How were the cisgender healthy volunteers recruited. Please specify.

At the meantime we are adding in the text also the information about how we recruited Volunteers. They were recruited in two territorial High Schools. Adolescents without psychopathological history were asked to complete the assessment questionnaires as volunteers after we presented the project and after the obtainment of written informed consent from parents and adolescents themselves.

Minor problems:

(1)Line 48: The authors wrote that the numbers of referrals to specialized gender clinics has increased. Just writing increased is very vague and not much informative. Please specify very shortly this trend in numbers (in general or for your own center). There have been papers published on the general trends. To give but one example: See Wiepjes et al. (2018). The Amsterdam cohort of gender dysphoria study (1972-2015). Journal of Sexual Medicine, 15: 582-590.

Unfortunately, we don’t have such data in our center because the Gender Dysphoria service of our clinic is still a young center. So, we are using general data and the work you suggested to be more detailed and less vague about this topic. We added in LINE 48 the following sentence: …As emerged in a cohort study on Gender Dysphoria where it is reported that in the Netherlands the prevalence was of 1:11,900 transwomen and 1:30,400 transmen in 1990 vs 1:2,800 transwomen and 1:5,200 transmen currently. Moreover, in adolescence, it is evident a progressive shift in the sex ratio from a higher proportion of male-assigned at birth to higher rates of female-assigned at birth, as shown in a study about sex ratio data where, comparing years 1988-2006 and 2007-2016 in an Amsterdam Clinic, the sex ratio changed significantly favoring female-assigned at birth (percentage of male-assigned at birth for the 2 time periods: 69.7% vs 46.8%).

(2) Line 55: "at rates higher". Please specify how much higher in numbers. At rates higher remains very vague. And specify if this is the case for only untreated adolescents with GD or also the case for treated adolescents (Compare for example with: Arnoldussen, M. et al. (2022). Self-perception of transgender adolescents after gender-affirming treatment: A follow-up into young adulthood. LGBT Health).

In LINES 55-56 we are adding the following sentence to be more detailed: …in fact, Connolly et al. in 2016 synthetized in a review the results of different studies of psychiatric comorbidities in adolescents with Gender Dysphoria, finding presence of depression in 12-64%, suicide attempts in 9-19%, self-Arm in 13-46%, eating disorders in 5-15%; moreover finding a significative difference (p<0,0001) in comparisons with Cisgenders for the presence of depression and history of suicide attempts; lastly this review included also two studies that compared for psychiatric symptoms, Gender Dysphoria adolescents social transitioned or treated with steroid suppression, finding no difference with Cisgender of the same age.

(3)Line 59: "we argued that that". Should that not be: We argue that.

Thanks, we corrected this line.

(4)Line 66: What do you mean with the construct "tendency to be attracted to life"?

We are sorry that maybe we used this construct in a too cryptical way. We are using a citation of Orbach 2001, to explain what we mean: “The body is a source of satisfaction and pleasure that enhances the tendency for life preservation and attraction to life and serves as a shield against self-destruction, while bodily dissatisfaction may increase suffering and intensify self-destructive attitudes”; and we are better characterizing this construct according to this citation.

(5)Line 83: "with a control group of healthy cisgender volunteers". Please replace control by comparison group. And  do so in the whole manuscript. 

Thanks, the manuscript is updated according with this suggestion.

(6)Line 93: "over a period of 24 months". Please specifiy the date when this period started, respectivelyended.

Thank you we specified the period in that started from April 2019 and ended in April 2021.

(7)Line 96 and 97: Why were Intellectual Disability and Autism Spectrum Disorder used as exclusion cirteria. Please specify. And why were other psychiatric syndromes (as for example bordeline personality disorder, or gilles de la tourette not used as exlusion criteria). Furthermore, please discuss in the general discussion the repercussions of the used exclusion criteria on the (lack of ?) representativeness of the sample.

Thank you for this useful request of specification about exclusion criteria of our study. We are adding more details about this topic in the lines 96 e 97. We excluded neurodevelopmental disorders (according to DSM-5) that could have had more influence on the dimensions examined in the study. In particular we excluded ID because it can compromise the validity of the scores of the questionnaires; moreover, we excluded ASD for two main reasons: the first is the frequent overlapping between Gender Dysphoria and Autism that could have represented a confounding factor; the second reason is that the way emotion regulation and emotional understanding acts in ASD patients have different mechanisms, different awareness and produce different behaviors in this patients. About the other neurodevelopmental disorders, we excluded the conditions that could have an influence on the examined dimensions. The incidence of the other pathologies has an impact on older ages, for this reason we believe that they haven’t a real incidence on the aims of our research.

(8)Line 100/101: Please specify in a concrete way what you mean by neurodevelopmental disorders and how they were differentiated from psychiatric symptoms.

About this specification we mean neurodevelopmental disorders according to DSM-5, so we are adding this detail in the text.

(9)Line 105 and 107: Psychopathological assessment. The reviewer suggest to use: Assessment of psychopathology instead of Psychopathological assessment.

Thanks, the manuscript is updated according with this suggestion.

(10)Line 109: SAFA: Is this an Italian scale or a scale translated in Italian? Are the reported Cronbach's alpha calculated on the original version or on the Italian data? Besides Chronbachs alpha, have there been studies validating this scale in other ways in the Italian version?

(11)For the other used scales the same question: Have there been studies validating the scales in other ways (for example divergent, convergent or predictive validity) using Italian data? 

Dear revisor we are answering to 10 and 11 comment together. The Cronbach’s alpha reported for SAFA is the one produced by validation and standardization of the original Italian test. About the other requests we upgraded the all of the paragraphs of the scales in methodology with more details and with more references about validations.

(12)Line 190-193: "both have a role" . What do the authors mean by both have a role. It seems they are suggesting causality, based on 1 measurement point. But, one measurement does not allow such causal interpretation. The same holds for the causal interpretations in the next paragraph. Furthermore, it seems that the authors are of the opinion that what applies for body image dissatisfaction applies also for body investment. Why, given that both constructs are not the same?

We are again sorry about this explicit causative inference. As said before, we provided a new version of the manuscript with a point in the limitation paragraph about this topic and traying to remove the causal relationship we explained in the text due to the cross-sectional nature of the study.

(13)Line: 207: What do the authors mean by maturation of body incongruence awareness? For example, why didn't they use the intensification of ...?

We provided to modify the manuscript with this useful suggestion, using the term “intensification of” instead of “maturation of” in the cited line because it is a more pertinent term.

(14)Line 235-238: The limitations of the study are discussed in a very superficial way. Please elaborate the limitations. To give one example: to what extent are the results influenced by the specific cultural context in which the research was done? To give another example: what are the limitations of the used in- and exclusion criteria? See also earlier made comments. And, specify also the strenghts of the study!

Thank you so much for this suggestion we totally reformed the paragraph of limitations following all of the interesting comments of your revisions, and we also added some lines about the strengths as you suggested.